# Effects of Neuromuscular Electrical Stimulation Combined with Exercises versus an Exercise Program on the Physical Characteristics and Functions of the Elderly: A Randomized Controlled Trial

**DOI:** 10.3390/ijerph18052463

**Published:** 2021-03-03

**Authors:** Eun Mi Jang, So Hyun Park

**Affiliations:** Department of Physical Therapy, Youngsan University, 288, Junam-ro, Yangsan-si, Gyeongnam-do 50510, Korea; gambare21@naver.com

**Keywords:** elderly, fall prevention, strengthening exercise, NMES with strengthening exercise

## Abstract

(1) Background—The application of neuromuscular electrical stimulation (NMES) combined with low-intensity exercise to the elderly can be more efficient than low-intensity exercise only in terms of delaying the loss of muscle mass. We aimed to assess the adjunct of NMES to low-intensity lower limb strengthening exercise to prevent falls in frail elderly for a relatively short period of 4 weeks. (2) Methods—Thirty elderly women aged 65 or above were randomly categorized into three groups: control group (CON, *n* = 8), exercise group (EX, *n* = 10), and NMES with exercise group (EX + NMES, *n* = 9). The exercise group took part in a lower limb strengthening exercise program for one hour three times a week for four weeks. Furthermore, the NMES with exercise group had added NMES stimulation when exercising. The limbs’ muscle mass, body fat mass, calf circumference, grip force, five times sit-to-stand test, timed up-and-go test (TUG), one-leg stand test, and Y-balance test (YBT) were evaluated at baseline and 4 weeks after. (3) Results—Comparisons between the three groups showed that the TUG was significantly decreased and the YB was significantly increased in NMES with exercise group (*p* < 0.05). (4) Conclusions—These results suggested that a combination of NMES stimulation and exercises was more helpful in strengthening balance than exercises alone in the short term.

## 1. Introduction

The elderly are characterized by a loss of muscle mass and strength, and as the aging progresses, the severe versions of this phenomenon lead to falls, fractures, and hospitalizations, especially in postmenopausal women [1]. Reduced strength, lack of participation in heavy outdoor work, and no habitual exercise make the elderly more fragile due to a greater loss of muscle mass and balance, which subsequently cause falls and fractures [2]. To break this vicious circle, it would be very useful to find an efficient method for preserving muscle mass and strength, even with low-intensity exercise, since the elderly have difficulty with high-intensity physical activity. 

Recently, neuromuscular electrical stimulation (NMES) has been used for improving muscle properties through repetitive muscle contractions [3]. Several researchers have reported on the NMES being an alternative, low-cost, efficient, and less painful approach [3,4,5,6]. Thus, the application of NMES during rehabilitation would improve exercise tolerance [5], effectively slowing muscle wasting during denervation or immobilization, increasing voluntary strength in partially paralyzed muscle [7,8], preventing skeletal muscle weakness [9], and optimizing the recovery of muscle strength [4]. 

Thus, we considered that the even short period of application of NMES combined with low-intensity exercise to the elderly can be more efficient than low-intensity exercise alone at delaying the loss of muscle mass, which may reduce the risk of falls [10,11]. 

To our knowledge, no studies have determined the effects of a short application period of NMES combined with exercise in the elderly. Previous study has not shown the effect of the addition of NMES to exercise [3]. Most studies on NMES primarily consisted of 6-to-12-week-long exercise programs [5,6,11]. A few studies conducted the program for 4 weeks but they did not observe the loss of muscle mass in the elderly [10,12]. 

Therefore, we aimed to evaluate the addition of NMES to lower limb strengthening exercises to prevent falls in the elderly for a relatively short period of 4 weeks. 

## 2. Materials and Methods

### 2.1. Study Participants

Participants were recruited from community-dwelling elderly females in Changwon-shi N town, South Korea. A total of 30 elderly women aged 65 or above were recruited using posters in the community. The inclusion criteria were as follows: able to perform exercise programs according to verbal instructions, no neurological disease, no difficulty in independent daily life, and the ability to display an unsupported gait in order to perform the evaluation and intervention program. The exclusion criteria were as follows: (a) lacked language skills for communication; (b) using a pacemaker to avoid overlapping the NMES electrical signals; (c) chronic disease, such as high blood pressure, diabetes, and heart disease, not controlled with medication; (d) difficulty in performing the intervention program due to musculoskeletal problems. The present study was approved by the Institutional Review Board of Youngsan University (No. YSUIRB-202006-HR-070-02). Informed consent was obtained from the participants. 

### 2.2. Determination of the Sample Size

For the sample size determination, the calf circumference became the primary endpoint in this study. The significance level (*α*) = 0.05, power (1 − *β*) = 0.85, effect size = 1, and a 10% drop out rate were used. A sample size of 9 was derived, yielding a total of 30 subjects being required. 

### 2.3. Procedure

Thirty people participated in the study, where they were randomly categorized into three groups: control group (CON), exercise group (EX), and NMES with exercise group (EX + NMES). During the experiment, two subjects in the CON group and one subject in the EX + NMES group were dropped out of the experiment because the subject herself was unable to continue the experiment. Therefore, 8 subjects in the CON group, 10 subjects in the EX group, and 9 subjects in the EX + NMES group participated in the experiment. The randomized controlled trial (RCT) study followed the consolidated standards of reporting trials (CONSORT) guidelines; the CONSORT flow chart is shown in Figure 1. The general characteristics of the subjects are presented in Table 1.

Exercise group: The exercise intervention was done three times a week for 4 weeks, with each session lasting one hour. The exercise intervention was divided into three parts; (1) stretching exercise of the upper and lower limb muscles, such as the trapezius, deltoid, triceps, latissimus dorsi, pectoralis major quadriceps femoris, gluteus maximus, gastrocnemius, and soleus muscles, during the first 10 min to warm up and prevent injury. (2) Lower limb muscle strengthening exercises were done to improve the strength and control of muscles around the hip, knee, and ankle joints for the next 40 min. The lower limb muscle strengthening exercises included a straight leg raise and bridge in a supine position, sidelying hip abduction, hip abduction and extension in a standing position, and heel raise and squat in a standing position. In addition, in a sitting position, the subjects performed knee extension, hip adduction, and sit-to-stand exercises. Participants initially performed one set of 10 repetitions for each exercise, which gradually increased to two sets of 10 repetitions (a total of 20 repetitions) for each exercise. (3) Cool down exercises, which were the same as the stretching exercises, were performed for the final 10 min for relaxation. 

NMES with exercise group: Participants in this group received NMES with the exercise for 4 weeks. The NMES with exercise program included the same methods with exercise group, 10 min of warm-up, 40 min of lower limb strengthening exercises, and 10 min of cooldown. Additionally, NMES was simultaneously applied. The NMES (LT1061, Supia, Korea) parameters were a frequency of 35 Hz and a duration of 300 μs, set as a square wave for 20 min to provoke muscle contraction. A pulse current (mA) was applied to elicit muscle contractions without pain or discomfort in either leg (average 10–12 mA). The electrodes were attached to the vastus medialis and vastus lateralis muscles of both thighs before the subject started the lower limb muscle strengthening exercise. The participants were instructed to perform the lower limb muscle strengthening exercise with the NMES stimulator attached to both thighs during the 20 min. Langeard et al. [12] reported a significant improvement in physical function when electric stimulation was provided three to four times a week for 20–30 min. Based on this, electrical stimulation for our participants was provided for 20 min. After NMES, lower extremity muscle strengthening exercises were performed for 20 min without NMES. 

Control group: To provide a baseline assessment, these subjects were allowed to avoid exercise during the experimental period and live as usual. 

Before and after intervention, all subjects performed an evaluation of the physical factors and the functional effects, such as calf circumference (CC), handgrip strength (HGS), sit-to-stand test (STS), timed up-and-go test (TUG), one-leg stand test (OLS), and the Y-balance test (YBT). All assessments were performed by the same geriatric physiotherapist who has experience with a number of elderly studies. 

### 2.4. Outcome Measurements

#### 2.4.1. Body Composition 

The use of bioelectrical impedance analysis (BIA) is widespread because of its inexpensiveness, ease of measurement, and quick assessment procedure for measuring body composition [13]. In Body 270 (IB270) (InBody USA, Cerritos, CA, USA) was used to measure the limb muscle mass and body fat mass. In order to increase the accuracy of the BIA results, the participants removed metallic accessories before the assessment and took off their socks. They stood on two metallic electrodes barefoot and grasped two metallic grip electrodes, one in each hand. Both arms were kept open such that they did not contact the torso. It was measured after having a bowel movement without eating breakfast.

#### 2.4.2. Calf Circumference

Measuring the calf circumference is simple assessment method and suggests that a lower CC is related to greater disability and lower physical function in elderly females [14]. Each subject was asked to stand with both feet spread shoulder-width apart. In this position, the circumference of the greatest girth of the calf was measured using an inelastic tape measure. 

#### 2.4.3. Handgrip Strength

Handgrip strength is used to measure the overall body muscle strength and physical function [15,16]. The intraclass correlation coefficient was recorded (ICC = 0.99) [17]. The handgrip strength was assessed using a hydraulic hand dynamometer (Baseline, Fabrication Enterprises Inc., Irvington, NY, USA). The grip strength of the dominant hand was measured with participants sitting upright on the chair and with the arm of the measured hand unsupported and parallel to the body. The measurements were repeated three times. Three minutes of rest were provided between each measurement. The average values were used for the analysis. 

#### 2.4.4. Five Times Sit-to-Stand Test

The five times sit-to-stand test is used to assesses functional lower limb muscle strength, balance control ability, and fall risk in older adults [18]. STS has demonstrated high test–retest reliability (ICC = 0.89) [19]. Each subject sat on a chair that was about 46 cm in height with their arms crossed in front of their chest. When the physiotherapist ordered the “start,” the time it took for the subject to get up from the chair five times as soon as possible was measured. The time was recorded with a stopwatch to the nearest 0.01 s. 

#### 2.4.5. Timed Up-and-Go Test

The timed up-and-go test was assessed to measure the functional mobility and dynamic balance ability. For the TUG, an excellent test–retest reliability has been calculated in older adults (ICC = 0.99) [20]. The TUG measured the time it took for the subject to stand up out of the chair with an armrest, turn at the return point that was 3 m away, and return to sit on the chair. The time was recorded with a stopwatch to the nearest 0.01 s. A lower time to perform the TUG represents that the participant has a good balance ability.

#### 2.4.6. One-Leg Stance Test

The one-leg stance test was measured to evaluate the static balance capability of the subject. The inter-rater reliability has been determined to be excellent (ICC = 0.99) [21]. Each subject stood with their feet shoulder-width apart. Then, at the same time as the instructor’s sign, the subject raised the non-dominant leg with the hip joint and knee joint held at 90°. The measurement used a digital stopwatch and was terminated if the raised leg was shaken to contact the ground again or the foot supporting the ground moved from its original position. In order to prevent falls, an assistant was allowed to stand next to the subject during the assessment. 

#### 2.4.7. Y-Balance Test

The Y-balance test was used for measuring dynamic balance capability. The decrease in YBT distance could be associated with reduced muscle strength in older females. The inter-rater reliability of the YBT is excellent (ICC = 0.95) [22]. The YBT measured the distance as the subject extended the opposite leg in the forward, posteromedial, and posterolateral directions while supporting the weight with one leg. They practiced six times in each direction with their hands on their waist. For normalization of the YBT distance, the length of the lower extremities from the anterior superior iliac spine to the medial malleolus was measured in a supine position. The standardization was calculated as percentages: (measured distance/leg length) × 100. The subject was evaluated three times in each direction and the mean value was used for the analysis. 

### 2.5. Statistical Analysis

Frequency analysis and technical statistics were used to identify the general characteristics of each group. The Shapiro–Wilk normality test and the parametric test were used to determine the differences between the groups in terms of the intervention. One-way ANOVA was used to identify the differences between the groups. Significant main effects were followed by Bonferroni’s post hoc procedures. A paired *t*-test was used to compare the dependent variables’ changes before and after the intervention within groups. The statistical processing used SPSS version 25.0 (IBM, Corp, Armonk, NY, USA) for Windows, and the statistical significance level for all analyses was *p* < 0.05.

## 3. Results

### 3.1. Changes in the Physical Characteristics before and after Intervention

The changes in the physical characteristics of the three groups before and after the intervention are shown in Table 2. In the EX and NMES + EX group, the calf circumference significantly increased (*p* < 0.05). 

### 3.2. Comparison of the Difference between the Physical Characteristics before and after According to the Intervention in the Three Groups

There was no significant difference in the skeletal muscle mass, body fat mass, calf circumference, and handgrip strength between each group (*p* > 0.05) (Table 3).

#### 3.2.1. Changes in Functional Effects before and after the Exercise Intervention

The changes in the functional effects of the three groups before and after the intervention are shown in Table 4. As a result of conducting the paired *t*-test, the EX group showed significant differences in other functional effects variables except for HGS and TUG (*p* < 0.05). The EX + NMES group showed statistically significant differences in all functional effect variables (*p* < 0.05).

#### 3.2.2. Comparisons of the Variables between Groups Regarding the Functional Effects 

The results of the one-way ANOVA showed significant differences in TUG and YBT (*p* < 0.05). In the Bonferroni post hoc test, both TUG and YBT showed significant differences between the CON and EX + NMES groups (*p* < 0.05) (Figure 2). 

## 4. Discussion

We measured the changes in the physical characteristics by assessing the limb muscle mass, body fat mass, calf circumference, and physical functional effects through the grip force, five times sit-to-stand test, timed up-and-go test, one-leg standing test, and Y-balance test at baseline and after 4 weeks. The results showed that the EX + NMES treatment was effective in increasing muscle circumference, similarly to the EX group, and excellent results in the functional effects of TUG and YBT.

We found a significant increase in the calf circumferences in the EX and EX + NMES groups. Moreover, there were no significant differences in the skeletal muscle mass, body fat mass, and calf circumference between the three groups after 4 weeks of intervention. A previous report [23] mentioned that NMES + exercise training during a 4-month intervention period performed at a low intensity improved the physical performance and muscle cross-sectional area. Another study [24] reported on the effects of high and low intensities of resistance training in the elderly for 52 weeks. The aforementioned training programs produced significant gains in thigh muscle strength, which was associated with fiber hypertrophy. In contrast, the effects of NMES applications for a shorter period on the physical characteristics are unclear. Moreover, a duration of 4 weeks is extremely short when it comes to changing the muscle and body fat masses. 

We found interesting results for the effects of the intervention on physical function. While the TUG produced significant decreases, the YBT results significantly increased in the EX + NMES group. Kim et al. [25] conducted a systematic review of the effects of NMES after an anterior cruciate ligament reconstruction. They reported that NMES might result in equal to moderately positive effects on quadricep strength 4 weeks after the operation compared to exercise alone or electromyographic feedback. Another systematic review [12] published similar results. The methodology was heterogeneous among the NMES studies, either targeting calf or thigh muscles. Moreover, they comprised studies of different duration and intensities. Despite this, proprioceptive processing may report effects on physical function [26]. Therefore, the combined application of exercise and NMES stimulation was found to be more effective in improving the physical function in the elderly within a short duration than exercise alone. 

The small sample size and relatively short study period were the major limitations. Four weeks is too short to change the physical characteristics. However, it is noteworthy that our results showed that short-term NMES application combined with exercise had the greatest effects on TUG and YBT. There was difficulty in the recruitment and continuing follow-up exercise in the participants. This could be attributed to the current pandemic. In addition, the electrical properties of NMES were not diversified; as such further randomized controlled trials are required to undertake measurements while considering the electrical properties. Our results are meaningful as they shed light on the short-term effects of exercise and NMES in the elderly

## 5. Conclusions

In the short term, a combination of exercise and electrical stimulation appears to be more helpful in strengthening balance in frail older women than exercise alone. The combination could have positive proprioceptive effects for preventing falls in the elderly but more research is needed.

## Figures and Tables

**Figure 1 ijerph-18-02463-f001:**
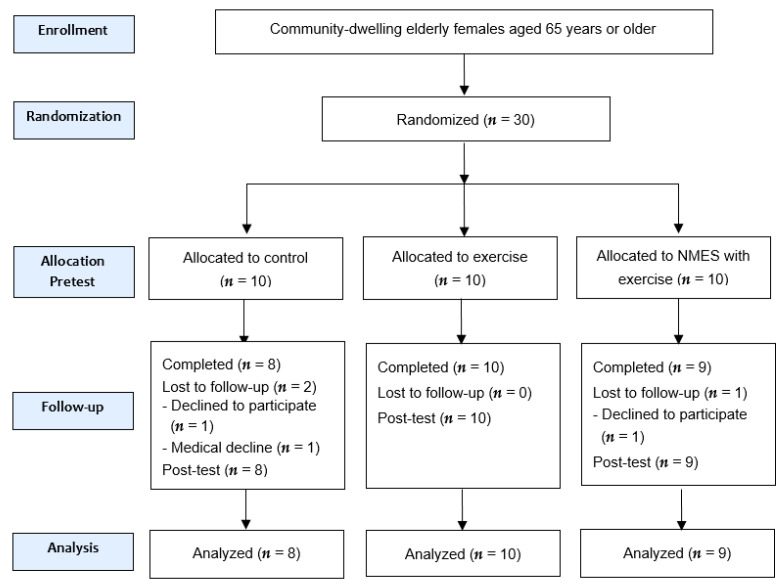
CONSORT diagram of participants. CONSORT: consolidated standards of reporting trials, NMES: neuromuscular electrical stimulation.

**Figure 2 ijerph-18-02463-f002:**
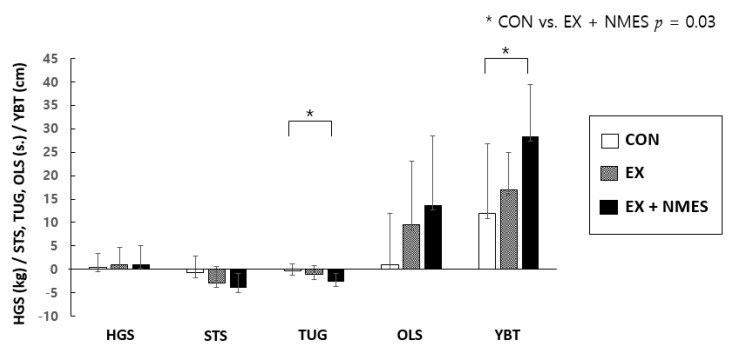
Comparisons of the difference in functional effects before and after the intervention in the three groups were indicated using means and standard deviations. To examine the relationships between the three groups, a one-way ANOVA analysis was conducted. The EX + NMES group improved in the TUG and YBT compared to the CON group. CON: control group, EX: exercise group, EX + NMES: NMES (neuromuscular electrical stimulation) with exercise group, HGS: handgrip strength, STS: sit-tostand test, TUG: timed up-and-go test, OLS: one-leg stance, YBT: Y-balance test.

**Table 1 ijerph-18-02463-t001:** General characteristics of the participants (*n* = 27).

Characteristics	CON Group (*n* = 8)	EX Group (*n* = 10)	EX + NMES Group (*n* = 9)	*p*
Age (years)	71.88 ± 6.69	73.3 ± 4.50	73.22 ± 4.76	0.83
Height (cm)	160.88 ± 6.08	154.46 ± 5.51	157.33 ± 4.74	0.06
Weight (kg)	59.88 ± 4.64	53.72 ± 5.40	53.69 ± 8.62	0.10

Values are expressed as mean ± standard deviation. CON: control, EX: exercise, NMES: neuromuscular electrical stimulation, EX + NMES: NMES with exercise.

**Table 2 ijerph-18-02463-t002:** Changes in the physical characteristics before and after the intervention (*n* = 27).

Variable	Group	Baseline	At 4 Weeks	*t*	*p*
Skeletal muscle mass (kg)	CON	22.34 ± 3.09 ^a^	22.13 ± 2.84	1.50	0.18
	EX	20.22 ± 2.14	19.95 ± 2.10	1.98	0.08
	EX + NMES	20.56 ± 2.45	20.43 ± 2.34	1.21	0.26
Body fat mass (kg)	CON	31.3 ± 4.50	31.88 ± 4.01	−1.40	0.20
	EX	29.31 ± 6.32	30.12 ± 5.30	−1.54	0.16
	EX + NMES	29.02 ± 5.20	28.86 ± 5.50	0.77	0.46
Calf circumference (cm)	CON	33.46 ± 0.96	33.69 ± 0.88	−1.00	0.35
	EX	31.89 ± 1.28	32.85 ± 0.85	−3.67	0.01 *
	EX + NMES	31.57 ± 2.98	32.61 ± 2.87	−3.90	0.01 *

^a^ Mean ± SD; * *p* < 0.05. CON: control, EX: exercise, NMES: neuromuscular electrical stimulation, EX + NMES: NMES with exercise.

**Table 3 ijerph-18-02463-t003:** Comparisons of variables among groups in physical characteristics (*n* = 27).

Variable	CON Group (*n* = 8)	EX Group (*n* = 10)	EX + NMES Group (*n* = 9)	*p*
Skeletal muscle mass (kg)	−0.21 ± 0.40 ^a^	−0.12 ± 0.30	−0.27 ± 0.43	0.71
Body fat mass (kg)	0.58 ± 1.16	0.81 ± 1.66	−0.17 ± 0.65	0.24
Calf circumference (cm)	0.23 ± 0.64	0.96 ± 0.83	1.04 ± 0.80	0.08

^a^ Mean ± SD. CON: control, EX: exercise, NMES: neuromuscular electrical stimulation, EX + NMES: NMES with exercise.

**Table 4 ijerph-18-02463-t004:** Changes in the functional effects before and after the intervention (*n* = 27).

Variables	Group	Baseline	At 4 Weeks	*t*	*p*
Handgrip strength (kg)	CON	20.78 ± 3.32 ^a^	21.19 ± 3.21	−0.41	0.10
	EX	19.71 ± 4.90	20.67 ± 3.93	−0.73	0.07
	EX + NMES	20.67 ± 5.29	21.81 ± 4.63	−0.88	0.04 *
Sit-to-stand (sec)	CON	14.26 ± 1.57	13.49 ± 3.10	0.59	0.57
	EX	12.83 ± 3.38	9.92 ± 1.49	2.57	0.03 *
	EX + NMES	13.27 ± 3.28	9.37 ± 2.25	3.98	0.004 **
Timed up-and-go test (sec)	CON	10.39 ± 1.25	10.10 ± 1.18	0.59	0.58
	EX	10.99 ± 1.55	9.87 ± 1.59	1.83	0.10
	EX + NMES	12.60 ± 2.56	9.88 ± 1.50	4.26	0.003 **
One-leg stance (sec)	CON	27.46 ± 18.52	28.51 ± 20.51	−0.27	0.79
	EX	26.27 ± 17.97	36.06 ± 24.17	−2.35	0.04 *
	EX + NMES	25.42 ± 17.11	39.10 ± 24.16	−2.76	0.03 *
Y-balance test (cm)	CON	69.08 ± 9.93	80.96 ± 14.26	−2.25	0.06
	EX	81.24 ± 4.45	98.17 ± 8.75	−4.38	0.002 **
	EX + NMES	62.93 ± 12.01	91.30 ± 13.76	−5.12	0.001 **

^a^ Mean ± SD. * *p* < 0.05, ** *p* < 0.01. CON: control, EX: exercise, NMES: neuromuscular electrical stimulati on, EX + NMES: NMES with exercise.

## Data Availability

The data presented in this study are available in insert article.

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
