# Peer review of "Effects of Neuromuscular Electrical Stimulation Combined with Exercises versus an Exercise Program on the Physical Characteristics and Functions of the Elderly: A Randomized Controlled Trial"

_ijerph, 2021, doi:10.3390/ijerph18052463_

Round 1
Reviewer 1 Report
Unfortunately, I apologize to the authors. But I found deep gaps in the methodology. I apologize for being abrupt in any comments. I hope it can be reviewed and considered in a different perspective.
First of all I would suggest making the title clearer because it speaks of the word elderly, but then the first word of the manuscript is Sarcopenia. you dissert on "Physical Characteristics and Functions", but then in the abstract there is a risk of falls. Better be clear right away.
Abstract
13. I suggest that the real background is not that NMES is safe. But that older people are frail and at risk of falling. Correct me if I'm wrong.
15. In the second sentence, the goal of using the two interventions in combination to reduce the risk of falls in the elderly in the short term should be emphasized
17. Focus on the type of study, then randomized controlled .. 3 arms (if there is no limit to the words add the trial code/ not local ethics committee code)
24. I suggest less clear-cut conclusions. Better "these results suggest / seem .." Unfortunately they are 3 groups of 10 on elderly subjects ..
Introduction
30-37. So define sarcopenia to have a more solid NMES background(?). Better to rephrase the title.
51-52. “Thus, we hypothesized that the application of NMES to the elderly can be a safe and 51 efficient assistive exercise method to delay the progression of sarcopenia and reduce the 52 risk of falls [14, 15].”
No. From how you designed the trial, this is not the goal. If you only assessed the impact of NMES, then the second group had to be provided: exercise + NMES sham. In this study, you are evaluating physiotherapy + NMES intervention versus physiotherapy alone. You are considering adding NMES to conventional exercise as an intervention ...
51-52 “to delay the progression of sarcopenia and reduce the 52 risk of falls”. This is really good. I suggest “functional” progression of sarcopenia.
57.. As already discussed, unfortunately the addition of NMES to physical exercise.
- 30 Females, better write in the abstract
Methods. CONSORT guidelines? Trial Registration? Sample size calculation?
Unfortunately it is difficult to judge in this way. It cannot be considered as an RCT. Better a Pilot study, at least the CONSORT should be drawn up. Not to mention the number of participants, too small to draw conclusions of this nature.
No inclusion criteria .. Do these participants have DMII? CKD?
Results
Physical characteristics? Why would the 1cm increase in circumference suggest anything about adding NMES exercising in a sample of 30 people?
Discussion
Perhaps it was better to add an functional scale (FIM, Barthel or SF-36 etc)
I find it difficult to argue. If the STS is significantly improved in the E + N and E alone versus control group, it is difficult at this point to explain why to add NMES.
204 Exact
215-217 It could have been a good starting point for the introduction to the purpose and reason of this study
Reviewer 2 Report
Dear authors,
congratulations to your interesting study. I have few remarks:
Abstract: it would be nice to put also between groups comparison in the abstract. Also, the same words as in the title should not appear in keywords.
Introduction: Did you run your study according to any hypothesis? Perhaps it would be valuable to put this information and your expectations in the end of your introduction.
Method: The description of study group is lacking imprtant data. How did you recruit your participants? Were these elderly living in the community or were they institutionalized? Were the subjects healthy elderly, without any medication? Could there be any contraindication to the use of NMES (e.g. cardiostimulator)? Did you evaluate the nutritional status of the subjects? Moreover, the exercise intensity is not described - this is problematic as it is not possible to replicate your study.
Table 1 - mean±SD?
row 76 - were allowed to avoid excessive exercise - does it mean that they could actually exercise even being in the control group?
87 - thr
165 - between after and before
Some parts of the manuscript are difficult to understand - 92-94, 109, 149-150, 184
Reviewer 3 Report
Introduction:
- While the introduction does set the stage for this study, it is not very compelling that NMES is a viable complementary therapy to improve function and reduce fall risk.
- What is the burden of sarcopenia, falls etc. in older adults. How significant a problem is this? Provide some epidemiological information please.
- What are the actual costs associated with sarcopenia, loss of function, and falls in older adults? While the authors briefly mention economic burden, they do not go into any specifics. It would help paint a picture of the gravity of the problem if they include some numbers.
Materials and Methods
- While a general description of the resistance training program is provided, there is not sufficient detail on workload (i.e. repetitions, sets, etc.). Please provide more details.
- Generally, this section contains several awkward sentences that confuse more than enlighten. This paper might benefit from a professional editor.
- For the each outcome measure, why were they selected? References would help to establish their validity in assessing related outcomes in older adults.
- Who conducted the assessments? Were they trained, and what was the training?
- For BIA, was hydration status kept consistent or even assessed?
Results
- Significant differences were found in calf circumference, balance, grip, sit-to-stand, and TUG. While I recognize that falls are related to loss of balance etc, because the authors did not track falls or even examine a more complete list of risk factors for falling, I would de-emphasize fall risk in this paper.
Discussion:
- Need to discuss more limitations.
- What are the longer-term implications? Describe next steps.
In general, I wonder how much impact this paper will have on the field. This paper needs some professional editing for grammar, word choices, syntax, and structure. Overall, the findings are expected with only 4-weeks of training. However, baring some major feasibility issues, the authors have demonstrated that NMES can be administered safely in older adults and that some improvements are possible.
I would want to see the paper again after the introduction, methods, and results, and discussion are revised.
Round 2
Reviewer 1 Report
I thank the authors for the substantial improvements to the manuscript but I must suggest further modifications. Above all to emphasize to the reader that adding NMES to the exercises does not have a physical but a functional rationale.
- Please, it’s a pilot study
27-29. Could I suggest this aim? "We aimed to assess the adjunct of NMES to low-intensity lower limb strengthening exercise to prevent falls in frail elderly for a relatively short period of 4 weeks."
37-38. I suggest these conclusions: These results suggested that a combination of exercises and NMES stimulation is more helpful in strengthening balance than exercises alone in the short-term.
46-50. It should be emphasized that the phenomenon is more acute in elderly women. So as to later justify the choice to enroll only female patients
53-58 I would also emphasize the proven use in other pathologies:
- Gabriel Ribeiro de Freitas, Camila Szpoganicz, Jocemar Ilha; Does Neuromuscular Electrical Stimulation Therapy Increase Voluntary Muscle Strength After Spinal Cord Injury? A Systematic Review. Top Spinal Cord Inj Rehabil1 January 2018; 24 (1): 6–17. doi: https://doi.org/10.1310/sci16-00048
- Maffiuletti, N.A., Roig, M., Karatzanos, E. et al.Neuromuscular electrical stimulation for preventing skeletal-muscle weakness and wasting in critically ill patients: a systematic review. BMC Med 11, 137 (2013). https://doi.org/10.1186/1741-7015-11-137
- Marotta, N., Demeco, A., Inzitari, M. T., Caruso, M. G., & Ammendolia, A. (2020). Neuromuscular electrical stimulation and shortwave diathermy in unrecovered Bell palsy: A randomized controlled study. Medicine, 99(8), e19152. https://doi.org/10.1097/MD.0000000000019152
63-68 Better to move these sentences into discussion. In this section it is enough to justify the research with the fact that there are few papers of this kind.
- I suggest this this aim: “Therefore, we aimed to evaluate the addition of NMES to lower limb strengthening exercises to prevent falls in the elderly for a relatively short period of 4 weeks.”
unfortunately the study is organized as a + b versus b versus control. Unfortunately we cannot compare "a" versus "b", but just adding "a" to a "b" program.
101-112. Great. Just add a sentence about following the CONSORT guidelines and a CONSORT FLOW CHART as figure.
- Please remove this sentence.
- Exact, This confirms the fact that from the point of view of physical characteristics the addition of NMES is not justified.
400. Remove the sentence with this one: “Despite this, proprioceptive processing may report effects on physical function” -> Mignardot JB, Deschamps T, Le Goff CG, Roumier FX, Duclay J, Martin A, Sixt M, Pousson M, Cornu C. Neuromuscular electrical stimulation leads to physiological gains enhancing postural balance in the pre-frail elderly. Physiol Rep. 2015 Jul;3(7):e12471. doi: 10.14814/phy2.12471. PMID: 26229006; PMCID: PMC4552546. - Conclusions
I suggest these:
In the short term, a combination of exercise and electrical stimulation appears to be more helpful in strengthening balance in frail older women than exercise alone. The combination could have positive proprioceptive effects for preventing falls in the elderly, but more research is needed.
Author Response
Please see the attachment

This manuscript is a resubmission of an earlier submission. The following is a list of the peer review reports and author responses from that submission.